# Comparative Effect of Antihypertensive Drugs in Improving Arterial Stiffness in Hypertensive Adults (RIGIPREV Study). A Protocol for Network Meta-Analysis

**DOI:** 10.3390/ijerph182413353

**Published:** 2021-12-18

**Authors:** Iván Cavero-Redondo, Alicia Saz-Lara, Luis García-Ortiz, Cristina Lugones-Sánchez, Blanca Notario-Pacheco, Leticia Gómez-Sánchez, Vicente Martínez-Vizcaíno, Manuel Ángel Gómez-Marcos

**Affiliations:** 1Health and Social Research Center, Universidad de Castilla-La Mancha, 16171 Cuenca, Spain; ivan.cavero@uclm.es (I.C.-R.); blanca.notario@uclm.es (B.N.-P.); Vicente.Martinez@uclm.es (V.M.-V.); 2Rehabilitation in Health Research Center (CIRES), Universidad de las Américas, Santiago 72819, Chile; 3Unidad de Investigación en Atención Primaria de Salamanca (APISAL), Instituto de Investigación Biomédica de Salamanca (IBSAL), Gerencia de Atención Primaria de Salamanca, Gerencia Regional de Salud de Castilla y León (SACyL), Avenida de Portugal 83, 37005 Salamanca, Spain; lgarciao@usal.es (L.G.-O.); crislugsa@gmail.com (C.L.-S.); leticiagmzsnchz@gmail.com (L.G.-S.); magomez@usal.es (M.Á.G.-M.); 4Departamento de Ciencias Biomédicas y del Diagnóstico, Universidad de Salamanca, Calle Alfonso X el Sabio s/n, 37007 Salamanca, Spain; 5Facultad de Ciencias de la Salud, Universidad Autónoma de Chile, Talca 3460000, Chile; 6Departamento de Medicina, Universidad de Salamanca, Calle Alfonso X el Sabio s/n, 37007 Salamanca, Spain

**Keywords:** hypertensive adults, arterial stiffness, antihypertensive drugs, network meta-analysis

## Abstract

(1) Background: Arterial stiffness is closely and bi-directionally related to hypertension and is understood as both a cause and a consequence of hypertension. Several studies suggest that antihypertensive drugs may reduce arterial stiffness. Therefore, effective prescription of antihypertensive drugs should consider both blood pressure and arterial stiffness. The aim of this protocol is to provide a review comparing the effects of different types of antihypertensive drug interventions on the reduction of arterial stiffness in hypertensive subjects. (2) Methods: The literature search will be performed through the MEDLINE, EMBASE, Cochrane Central Register of Controlled Trials, Cochrane Database of Systematic Reviews, and Web of Science databases. Randomised clinical trials assessing the effect of antihypertensive drug interventions on arterial stiffness measured in subjects with hypertension will be included. A frequentist network meta-analysis will be performed to determine the comparative effects of different antihypertensive drugs. (3) Results: The findings of this study will be published in a peer-reviewed journal. (4) Conclusions: This study will provide evidence for health care professionals on the efficacy of different antihypertensive drugs in decreasing arterial stiffness; in addition, it will analyse the efficacy of the drugs not only in terms of arterial stiffness but also in terms of blood pressure treatment.

## 1. Introduction

Hypertension is the leading cause of mortality in adults and affects 66% of those over 60 years of age [1]. Hypertension is important because it is a major risk factor for cardiovascular diseases (CVDs), such as heart failure, coronary heart disease, peripheral artery disease, or stroke [2]. Furthermore, considering the progressive ageing of the population and the increasing prevalence of hypertension with age, it is plausible that the clinical importance of hypertension will be greater in the coming years [3].

To maintain normal blood pressure (BP), physiological arterial elasticity is important [4]. Loss of elasticity, or arterial stiffness (AS), is due to changes in arterial wall structure and function that occur physiologically with ageing but may be accelerated, in addition to genetic determinants, by a variety of other factors such as obesity, insulin resistance, and diabetes [5]. Pulse wave velocity (PWV) is the universally accepted measure for AS [6], and a 1 m/s increase in PWV is considered to increase the risk of suffering a cardiovascular event by 14% and that of dying from CVD by 15% [7].

Different studies have shown that AS is involved in both the pathogenesis and prognosis of hypertension [8,9,10], establishing an association between AS and BP levels, as increased BP is an important cardiovascular risk factor leading to arterial wall damage [11]. Given that hypertension affects more than half of people over 50 years of age in industrialized countries and is responsible for 51% of deaths from stroke and 45% of deaths from heart disease, in the last three decades, we have seen major advances in the treatment of hypertension, with new antihypertensive drugs being incorporated into the therapeutic arsenal to lower BP and improve AS [12,13,14,15].

However, although physicians are well aware of the effects of different antihypertensive drugs in lowering BP, the effects of these drugs on AS do not seem to be considered, despite evidence supporting AS as an independent risk factor for CVD and overall mortality [16]. Therefore, the aim of this protocol is to synthesize and assess the available scientific evidence on the efficacy of antihypertensive drugs on AS in patients with hypertension.

## 2. Materials and Methods

### 2.1. Protocol Register

This protocol for a network meta-analysis followed the Preferred Reporting Items for Systematic Review and Meta-Analysis Protocols (PRISMA-P) [17] and the Cochrane Collaboration Handbook [18]. Additionally, this protocol for a network meta-analysis has been registered in PROSPERO (CRD42021276360).

### 2.2. Ethics

Since no primary data will be collected from patients with hypertension, no ethics committee approval will be required for this study

### 2.3. Inclusion/Exclusion Criteria

#### 2.3.1. Type of Studies

RCTs will be included without language restrictions.

#### 2.3.2. Type of Participants

Studies evaluating the effect of different antihypertensive drugs on the reduction of AS and systolic blood pressure (SBP) or pulse pressure (PP) in hypertensive adults with a primary diagnosis of hypertension according to the diagnostic criteria of the International Classification of Diseases (ICD-11) (>18 years of age and of both genders) will be selected. If two or more studies provide data on the same sample, the one that presents the most detailed results or provides the largest sample size will be chosen.

#### 2.3.3. Intervention Types

Studies using any of the different drugs in the antihypertensive groups as an intervention (see the list of drugs in Appendix A), as well as possible drug combinations, will be suitable for inclusion, as will studies comparing different types of antihypertensive drugs and examining antihypertensive treatment with or without a control group. However, studies combining antihypertensive drugs with nutritional or lifestyle interventions will be excluded when data regarding the effect of antihypertensive drug interventions on AS cannot be extracted separately.

#### 2.3.4. Outcome Assessment Type

Reduction of AS parameters will be measured as primary outcomes: PWV, the augmentation index (AIx), the ambulatory AS index (AASI) and the cardio-ankle vascular index (CAVI). As a secondary outcome measure, the effect on SBP and PP will be explored.

### 2.4. Search Methods for Study Identification

#### Electronic Search

The bibliographic search will be performed through the MEDLINE, EMBASE, Cochrane Central Register of Controlled Trials, Cochrane Database of Systematic Reviews, and Web of Science databases. The above searches will be supplemented by manual searches of published or ongoing RCTs in international trial registries (ClinicalTrials.gov) and on drug approval agency websites. Before the final analyses, the searches will be repeated just to include all current and potential studies. The studies found in the search will be managed through the Mendeley reference manager.

To perform the bibliographic search, the following search terms will be used in combination applying the Boolean operators: “hypertensive adults”, “hypertensive population”, “hypertensive subjects”, “arterial hypertension”, “antihypertensive treatment”, “antihypertensive drugs”, “beta-blockers”, acebutolol, atenolol, atenolol, betaxolol, bisoprolol, carteolol, esmolol, metoprolol, nadolol, oxprenolol, penbutolol, propranolol, timolol, celiprolol, carvedilol, labetalol, nebivolol, pindololol, diuretics, furosemide, bumetanide, torsemide, Bendroflumethiazide, chlorothiazide, chlorthalidone, hydrochlorothiazide, indapamide, polythiazide, trichlormethiazide, amiloride, eplerenone, spironolactone, triamterene, “angiotensin-converting enzyme inhibitors”, benazepril, captopril, cilazapril, enalapril, fosinopril, imidapril, lisinopril, moexipril, perindopril, quinapril, ramipril, trandolapril, zofenopril, “angiotensin II receptor antagonists”, candesartan, eprosartan, irbesartan, losartan, olmesartan, telmisartan, valsartan, “calcium channel blockers”, diltiazem, verapamil, amlodipine, felodipine, isradipine, lacidipine, lercanidipine, manidipine, nicardipine, “renin inhibitors”, aliskiren, “alpha-adrenergic receptor antagonists”, doxazosin, prazosin, terazosin, “centrally acting agents”, clonidine, methyl-dopa, rilmenidine, “direct acting vasodilators”, hydralazine, minoxidine, “arterial stiffness”, “pulse wave velocity”, PWV, “augmentation index”, Aix, “ambulatory arterial stiffness index”, AASI, “cardio-ankle vascular index”, CAVI, “randomised controlled trial”, “randomized clinical trial”, and RCT (Table 1).

### 2.5. Data Collection and Analysis

#### 2.5.1. Study Selection

Following the search, to screen eligible studies based on the inclusion criteria, the title and abstract will be assessed separately by two reviewers. The full text of the identified studies will be examined. Finally, two reviewers will verify the reasons why the studies were included or excluded (Figure 1). Reviewers will not disclose the following information: authors, institutions, or journals of the articles reviewed. Disagreements will be solved by consensus or with the intervention of a third researcher.

The following information on the included studies will be provided independently by two authors: (1) reference (first author and publication year); (2) country in which the study data were collected; (3) population characteristics (sample size, mean age, status (hypertension or uncontrolled hypertension)); (4) intervention characteristics (type of antihypertensive drugs (see the list of drugs in Appendix A), dose administered and frequency, length of treatment); (5) outcome (AS parameter (PWV, Aix, AASI, CAVI), measurement device, baseline levels) (Table 2). When necessary to obtain missing information from the studies, the corresponding author will be contacted.

#### 2.5.2. Assessing the Risk of Bias in Included Studies

Based on the recommendations of the Cochrane Collaboration Handbook, two authors will independently conduct a quality assessment [18]. Disagreements will be resolved by consensus or with the intervention of a third researcher.

The risk of bias of RCTs will be assessed using the Cochrane Collaboration’s tool for assessing risk of bias (RoB2) [19], according to six domains: selection bias, performance bias, detection bias, attrition bias, reporting bias, and other biases. The overall bias is considered “low risk of bias” when all domains are evaluated as “low risk”, “some concerns” when there is at least one domain evaluated as “some concerns”, and “high risk of bias” when there is at least one domain evaluated as “high risk” or when several domains are evaluated as “some concerns”.

#### 2.5.3. Grading the Quality of Evidence

We will use the Grading of Recommendations, Assessment, Development and Evaluation (GRADE) tool to assess the evidence quality and provide recommendations [20]. The GRADE tool includes the following five distinct steps for each outcome: (1) allocate an a priori classification of “high” to RCTs and “low” to observational studies; (2) “downgrade” or “upgrade” the initial rating based on: risk of bias, inconsistency, indirect evidence, imprecision, publication bias, large effect, dose–response relationship and all plausible biases that only reducing an apparent treatment effect; (3) allocate the final rating of the quality of evidence as “high”, “moderate”, “low”, or “very low”; (4) address other influencing factors that affect the recommendation strength of a course of action; (5) make a “strong” or “weak” recommendation [21].

### 2.6. Synthesis of Data

We will qualitatively summarize the included RCTs in an ad hoc table describing the direct and indirect comparisons. For each direct comparison between two interventions, a standard meta-analysis will be performed using the DerSimonian–Laird random-effects method [22]. We will assess heterogeneity using the *I*^2^ statistic [23], ranging from 0% to 100%. Based on the values of *I*^2^, we will categorize heterogeneity as not important (0% to 30%), moderate (30% to 60%), substantial (60% to 75%), or considerable (75% to 100%). We will consider the corresponding *p* values.

Sensitivity analysis will be conducted to evaluate the robustness of the pooled estimates, a reanalysis will be conducted by eliminating one study at a time.

Subgroup analyses will be conducted based on smoking status (non-smoker, ex-smoker, smoker) and type of PWv (central or peripheral PWv).

Random-effects meta-regression analyses will be approached to analyse whether mean age, sex, number of comorbidities, number of drugs beyond hypertensive agents, and duration of treatment changed the effect of antihypertensives drugs on AS.

Publication bias will be tested using Egger’s regression asymmetry test [24], setting a level of <0.10 to determine the presence of publication bias might be present.

A frequentist network meta-analysis will be performed to determine the comparative effects of different antihypertensive drug interventions. The effects of each intervention will be combined by Markov chain Monte Carlo frequentist methods [25] using STATA 15 (StataCorp, College Station, TX). Additionally, we will perform a Bayesian network meta-analysis as a sensitivity analysis using the gemtc 0.8–2 and BUGSNET packages in R (version 4.0.2).

The probability that each intervention is the most effective will be presented by rankograms. In addition, for each intervention, we will estimate the area under cumulative ranking (SUCRA) [26]. With SUCRA, a value between 0 and 1 is assigned to rank each intervention in the rankogram. A SUCRA value of approximately 1 will be the best intervention, and a SUCRA value of approximately 0 will be the worst intervention. SUCRA simplifies the information on the effect of each treatment into a single value, and all complex results of network meta-analysis are expressed with a few numbers. The SUCRA result is most meaningful when the difference in preference between consecutive ranks remains the same over the entire rating scale.

## 3. Results

The results of this research will be presented in a peer-reviewed journal.

## 4. Discussion

Hypertension and AD are closely related, and it is currently under debate whether hypertension is a cause or a consequence of AD [4,9,11,27]. Hypertensive subjects present a vascular phenotype characterized by endothelial dysfunction, vascular inflammation, arterial remodelling with loss of arterial distensibility and changes in collagen/elastin ratio, vascular muscle tone, transmural distending pressure, pro-inflammatory responses, and oxidative stress [28]. The importance of these structural changes lies in the fact that they produce a progressive reduction in arterial distensibility and elasticity, leading to an increase in AS [9,28].

An increase in AS leads to a higher blood flow velocity, which, in the long term, increases the risk of alterations in target organs such as the brain, kidneys and heart. Thus, AS is considered an independent risk factor for CVD in the general population, in the elderly population and in people with hypertension [6]. Therefore, to maintain a normal BP, physiological arterial elasticity is important [9].

There are numerous strategies for the treatment of hypertension [13]. Moreover, studies suggest that antihypertensive drugs could reduce AS by two mechanisms: first, by lowering SBP and thus reducing the mechanical stress of cardiac embolism on the walls of large arteries, and second, by improving the collagen–elastin ratio of the arterial wall, as some antihypertensive drugs modify the structure of the arterial wall [29,30]. In addition, the evidence available through various meta-analyses suggests that not all antihypertensive drugs are equally effective in improving AS even if they have similar efficacy in improving BP since their effects on arterial wall structure might be different [11]. Therefore, in light of current knowledge, the selection of the best antihypertensive treatment strategy should consider both aspects of a drug’s effect: BP lowering and AS improvement.

The objective of this study is to facilitate the protocol methodology for a new network meta-analysis of the effect of different antihypertensive drug types on AS in the hypertensive population to decrease the risk of CVD. In addition, this study intends to provide evidence that can be implemented in guidelines of clinical practice to recommend the best antihypertensive treatment for hypertensive patients.

## 5. Conclusions

With this study, we will provide evidence for health professionals on the efficacy of different antihypertensive drugs in lowering AS in addition to BP. Furthermore, considering the progressive ageing of the population and the increasing prevalence of hypertension with age, it is plausible that the clinical importance of hypertension will be greater in the coming years. For all these reasons, control of hypertension is a challenge of great interest for health care systems.

## Figures and Tables

**Figure 1 ijerph-18-13353-f001:**
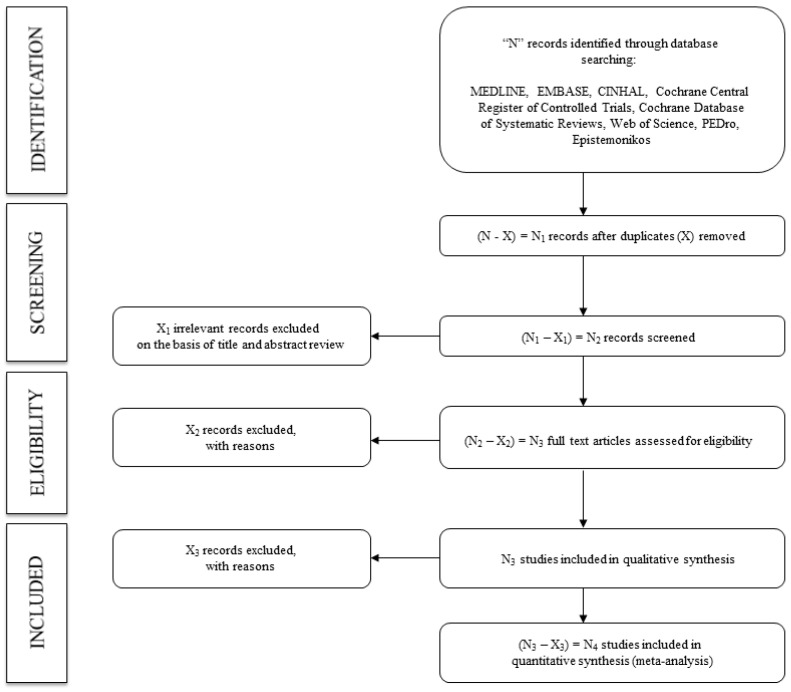
Flowchart of identification, screening, eligibility, and inclusion of studies.

**Table 1 ijerph-18-13353-t001:** MEDLINE search strategy.

“Hypertensive adults”. OR “Hypertensive population” OR “Hypertensive subjects” OR “Arterial hypertension”	AND	“Antihypertensive treatment” OR “Antihypertensive drugs” OR “Beta-blockers” OR acebutolol OR atenolol OR atenolol OR betaxolol OR bisoprolol OR carteolol OR esmolol OR metoprolol OR nadolol OR oxprenolol OR penbutolol OR propranolol OR timolol OR celiprolol OR carvedilol OR labetalol OR nebivolol OR pindololol OR Diuretics OR furosemide OR bumetanide OR torsemide OR bendroflumethiazide OR chlorothiazide OR chlorthalidone OR hydrochlorothiazide OR indapamide OR polythiazide OR trichlormethiazide OR amiloride OR eplerenone OR spironolactone OR triamterene OR “Angiotensin-converting enzyme inhibitors” OR benazepril OR captopril OR cilazapril OR enalapril OR fosinopril OR imidapril OR lisinopril OR moexipril OR perindopril OR quinapril OR ramipril OR trandolapril OR zofenopril OR “Angiotensin II receptor antagonists” OR candesartan OR eprosartan OR irbesartan OR losartan OR olmesartan OR telmisartan OR valsartan OR “Calcium channel blockers” OR diltiazem OR verapamil OR amlodipine OR felodipine OR isradipine OR lacidipine OR lercanidipine OR manidipine OR nicardipine OR “Renin inhibitors” OR aliskiren OR “Alpha-adrenergic receptor antagonists” OR doxazosin OR prazosin OR terazosin OR “Centrally acting agents” OR clonidine OR methyl-dopa OR rilmenidine OR “Direct acting vasodilators” OR hydralazine OR minoxidine	AND	“Arterial stiffness” OR “Pulse wave velocity” OR PWV OR “Augmentation index” OR Aix OR “Ambulatory arterial stiffness index” OR AASI OR “Cardio-ankle vascular index” OR CAVI	AND	“Randomised controlled trial” OR “Randomized clinical trial” OR RCT

**Table 2 ijerph-18-13353-t002:** Characteristics of studies included.

		Population Characteristics			Intervention Characteristics			Outcome		
Reference	Country	Sample Size	Mean Age	Status	Type of Antihypertensive Drugs	Dose	Length	Arterial Stiffness Parameter	Measurement Device	Baseline Levels
First author and year of publication	Country in which the study data were collected	Number of participants and percentage of female	Age (years) of the participants range or mean ± SD	Hypertension or Uncontrolled hypertension	Antihypertensive drugs included in the list of drugs in Appendix A	Dose administered and frequency	Length (months) of treatment	Type of arterial stiffness parameter (PWV, Aix, AASI, CAVI)	Arterial stiffness parameter measuring device	Baseline levels of the measured arterial stiffness parameter

Aix: augmentation index; AASI: ambulatory arterial stiffness index; CAVI: cardio-ankle vascular index; PWV: pulse wave velocity.

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
