# Peer review of "Comparative Effect of Antihypertensive Drugs in Improving Arterial Stiffness in Hypertensive Adults (RIGIPREV Study). A Protocol for Network Meta-Analysis"

_ijerph, 2021, doi:10.3390/ijerph182413353_

Round 1
Reviewer 1 Report
The study of Iván Cavero-Redondo et al., entitled “Comparative effect of antihypertensive drugs in improving arterial stiffness in hypertensive adults (RIGIPREV study). A protocol for network meta-analysis” highlights the clinical importance of hypertension, since this pathological state is a major risk factor for cardiovascular diseases (CVDs), and provides evidence for health professionals on the efficacy of different antihypertensive drugs in decreasing arterial stiffness, as well as in terms of blood pressure treatments. It is a well-structured and informative article presenting a protocol for network meta-analysis, which can determine the comparative effects of different antihypertensive drug interventions. However, I have a few minor concerns:
- I believe that the authors should incorporate in their manuscript a brief description of GRADE tool, Egger’s test and SUCRA tool, regarding the way they perform the analysis and their output.
- The manuscript contains a few syntax errors that must be corrected.
Author Response
Reviewer 1
The study of Iván Cavero-Redondo et al., entitled “Comparative effect of antihypertensive drugs in improving arterial stiffness in hypertensive adults (RIGIPREV study). A protocol for network meta-analysis” highlights the clinical importance of hypertension, since this pathological state is a major risk factor for cardiovascular diseases (CVDs), and provides evidence for health professionals on the efficacy of different antihypertensive drugs in decreasing arterial stiffness, as well as in terms of blood pressure treatments. It is a well-structured and informative article presenting a protocol for network meta-analysis, which can determine the comparative effects of different antihypertensive drug interventions.
Authors:
Thank you for the reviewer´s comment. We greatly appreciate the time the reviewer spent reviewing the manuscript.
However, I have a few minor concerns:
- I believe that the authors should incorporate in their manuscript a brief description of GRADE tool, Egger’s test and SUCRA tool, regarding the way they perform the analysis and their output.
Authors:
Thank you for the reviewer’s comment. As suggested, we have included a brief description of GRADE tool, Egger’s test and SUCRA tool.
In the methods section:
“ […]. The GRADE tool includes the following five distinct steps for each outcome: 1) Assign an a priori ranking of “high” to randomized controlled trials and “low” to observational studies; 2) “Downgrade” or “upgrade” the initial rating based on: risk of bias, inconsistency, indirect evidence, imprecision, publication bias, large effect, dose-response relationship and all plausible biases that only reducing an apparent treatment effect; 3) Assign the final grade of the quality of evidence as “high”, “moderate”, “low”, or “very low”; 4) Consider other factors that impact the strength of recommendation for a course of action; and 5) Make a “strong” or “weak” recommendation [21].”
“ […]. Additionally, publication bias will be assessed using Egger’s regression asymmetry test [23]. A level of <0.10 will be used to determine whether publication bias might be present.”
“ […]. SUCRA has the advantage that is simplifies the information on the effect of each treatment into a single number, and all the complex results of network meta-analysis are expressed with a few numbers. SUCRA is most meaningful when the difference in preference between successive ranks remains the same over the entire ranking scale. In the absence of such an interval scale, an otherwise attractive and simple SUCRA value can be misleading. Moreover, it is impossible to know what constitutes a modest or large difference in SUCRA between two treatments, either statistically or clinically.”
- The manuscript contains a few syntax errors that must be corrected.
Authors:
Thank you for the reviewer’s comment. As suggested, we have thoroughly reviewed the manuscript.

Reviewer 2 Report
Only comment is to add more search terms to make sure that you will capture all the relevant studies.
Author Response
Reviewer 2
Only comment is to add more search terms to make sure that you will capture all the relevant studies.
Authors:
Thank you for the reviewer´s comment. We greatly appreciate the time the reviewer spent reviewing the manuscript. As suggested, we have included the following search terms to capture all the relevant studies.
In the methods section:
“To perform the bibliographic search, the following search terms will be combined using Boolean operators: “hypertensive adults”, “hypertensive population”, “hypertensive subjects”, “arterial hypertension”, “antihypertensive treatment”, “antihypertensive drugs”, “beta-blockers”, acebutolol, atenolol, atenolol, betaxolol, bisoprolol, carteolol, esmolol, metoprolol, nadolol, oxprenolol, penbutolol, propranolol, timolol, celiprolol, carvedilol, labetalol, nebivolol, pindololol, diuretics, furosemide, bumetanide, torsemide, bendroflumethiazide, chlorothiazide, chlorthalidone, hydrochlorothiazide, indapamide, polythiazide, trichlormethiazide, amiloride, eplerenone, spironolactone, triamterene, “angiotensin-converting enzyme inhibitors”, benazepril, captopril, cilazapril, enalapril, fosinopril, imidapril, lisinopril, moexipril, perindopril, quinapril, ramipril, trandolapril, zofenopril, “angiotensin II receptor antagonists”, candesartan, eprosartan, irbesartan, losartan, olmesartan, telmisartan, valsartan, “calcium channel blockers”, diltiazem, verapamil, amlodipine, felodipine, isradipine, lacidipine, lercanidipine, manidipine, nicardipine, “renin inhibitors”, aliskiren, “alpha-adrenergic receptor antagonists”, doxazosin, prazosin, terazosin, “centrally acting agents”, clonidine, methyl-dopa, rilmenidine, “direct acting vasodilators”, hydralazine, minoxidine, “arterial stiffness”, “pulse wave velocity”, PWV, “augmentation index”, Aix, “ambulatory arterial stiffness index”, AASI, “cardio-ankle vascular index”, CAVI, “randomised controlled trial”, “randomized clinical trial”, and RCT (Table 1).”

Reviewer 3 Report
Cavero-Redondo et al., suggested a clinical protocol for network meta-analysis to compare the blood pressure lowering and the arterial stiffness reducing effects of frequently used antihypertensive drugs in different pharmacological families.
- It could be interesting to use the keyword "arterial hypertension" in the Medline search if the authors aim at investigating mainly this form of hypertension and not secondary hypertension forms.
- In Table 1, renin inhibitors are mentioned; however, they are not included in appendix A.
- In many cases, more antihypertensive drugs are combined in clinical use. How could the authors decide if a single drug or the combination has AS lowering effect in the case of clinical studies using more drugs?
- Do you plan to expand the characteristics (e.g., smoking habits, comorbidities, drugs beyond hypertensive agents, etc.) of studies included in the meta-analysis?
Author Response
Reviewer 3
Cavero-Redondo et al., suggested a clinical protocol for network meta-analysis to compare the blood pressure lowering and the arterial stiffness reducing effects of frequently used antihypertensive drugs in different pharmacological families.
Authors:
Thank you for the reviewer´s comment. We greatly appreciate the time the reviewer spent reviewing the manuscript.
- It could be interesting to use the keyword "arterial hypertension" in the Medline search if the authors aim at investigating mainly this form of hypertension and not secondary hypertension forms.
Authors:
The reviewer´s comment seems judicious. As suggested, we have included the keyword "arterial hypertension" in the Medline search.
- In Table 1, renin inhibitors are mentioned; however, they are not included in appendix A.
Authors:
Thank you for the reviewer’s comment. As suggested, we have included the keyword "renin inhibitors" in appendix A.
- In many cases, more antihypertensive drugs are combined in clinical use. How could the authors decide if a single drug or the combination has AS lowering effect in the case of clinical studies using more drugs?
Authors:
The reviewer´s comment seems judicious. As suggested, we have included studies involving drug combinations as inclusion criteria. Network meta-analysis allows us to compare (including single-drug treatments or drug combinations) and to know which intervention is the most effective in reducing arterial stiffness.
In the methods section:
“Studies using any of the different drugs in the antihypertensive groups as an intervention (see the list of drugs in Appendix 1) as well as possible drug combinations will be suitable for inclusion […].”
- Do you plan to expand the characteristics (e.g., smoking habits, comorbidities, drugs beyond hypertensive agents, etc.) of studies included in the meta-analysis?
Authors:
Thank you for the reviewer’s comment. As suggested, we have included sensitivity analysis, subgroup analyses and meta-regressions to assess possible variables that could influence the results, such as: age, sex, smoking status, comorbidities, drugs beyond hypertensive agents, type of pulse wave velocity and duration of treatment.
In the methods section:
“[…]. A sensitivity analysis (systematic reanalysis removing studies one at a time) was performed to assess the robustness of the summary estimates. Subgroup analyses were performed according to smoking status (non-smoker, ex-smoker, smoker) and type of PWv (central or peripheral PWv). Random effects meta-regression analyses addressed whether mean age, sex, number of comorbidities, number of drugs beyond hypertensive agents and duration of treatment, as continuous variables, modified the effect of antihypertensives drugs on AS […].”

Round 2
Reviewer 3 Report
This reviewer is satisfied with the answers and the modification of the MS.